# Laughter influences social bonding but not prosocial generosity to friends and strangers

**R. I. M. Dunbar**📧[1]*, **Anna Frangou**[2], **Felix Grainger**[1], **Eiluned Pearce**[1,3]

**1** Department of Experimental Psychology, University of Oxford, Oxford, United Kingdom, **2** Big Data Institute, University of Oxford, Oxford, United Kingdom, **3** Department of Psychiatry, University College London, London, United Kingdom

* robin.dunbar@psy.ox.ac.uk

## Abstract

Humans deploy a number of specific behaviours for forming social bonds, one of which is laughter. However, two questions have not yet been investigated with respect to laughter: (1) Does laughter increase the sense of bonding to those with whom we laugh? and (2) Does laughter facilitate prosocial generosity? Using changes in pain threshold as a proxy for endorphin upregulation in the brain and a standard economic game (the Dictator Game) as an assay of prosociality, we show that laughter does trigger the endorphin system and, through that, seems to enhance social bonding, but it does not reliably influence donations to others. This suggests that social bonding and prosociality may operate via different mechanisms, or on different time scales, and relate to different functional objectives.

## Introduction

Human societies are characterised by high levels of prosociality, both in the sense of acting generously (altruistically) towards others and in the sense of cooperating with others on some communal task. Such behaviour is often viewed as being a generalised predisposition, with a prominent role for oxytocin [1–6]. In fact, altruism and cooperation are not distributed indiscriminately, but are directed disproportionately towards members of one's own community and, especially, to close friends and family [7–12], suggesting that the prior existence of close social bonds may play an important role.

In primates in general, social bonds are underpinned by social grooming, and are mediated by β-endorphin upregulation in the brain [13–15] via the highly specialised afferent C-tactile peripheral neuron system that responds only to light slow stroking [16–18]. It is the endorphin system that provides the psychopharmacological underpinnings for social bonding [19–24] in a way that creates stable groups out of dyadic 'friendships' [25–27]. These bonded relationships are what allow primates to cooperate with, and behave altruistically towards, each other. This reflects a crucial contrast in the way the oxytocin and endorphin systems function: the one is endogenous (my oxytocin levels predispose me to behave prosocially towards you, but I cannot influence your behaviour) while the other acts exogenously (I can influence your β-endorphin levels directly and so make you be more prosocial towards me). In addition, endorphins have a serum half-life that is measured in hours [28], whereas that for oxytocin is measured in minutes [29–32], with half-life in the CNS being longer but proportionately similar [28, 33].

**Data Availability Statement:** Data included as ESM files.

**Funding:** (1) RIMD: School of Anthropology and Museum Ethnography [no grant numbers] (2) RIMD: Department of Experimental Psychology

The funders had no role in study design, data collection and analysis, decision to publish, or preparation of the manuscript.

**Competing interests:** The authors have declared that no competing interests exist.

Social grooming continues to function in humans just as it does in nonhuman primates [15, 34, 35]. However, increased demand for bonding larger numbers of individuals when time to do so is severely limited has resulted in grooming being augmented by a variety of other behaviours that also trigger the endorphin system without need for physical contact, thereby allowing more individuals to be 'groomed' simultaneously [36, 37]. These behaviours include laughter [38, 39], singing [40], dancing [41, 42], emotional storytelling [43]), feasting (social eating and alcohol consumption [44–46] and the rituals of religion [47], all of which elevate pain thresholds and/or endorphin uptake. Of these, all but laughter have also been shown to increase the sense of social bonding with specific individuals [40, 43, 47–50]. None, however, have been directly tested to determine whether they also influence the likelihood of prosociality (generosity to others).

The key issue here is whether social bonding behaviours like laughter trigger cooperation/ generosity directly or whether generosity is the outcome of an intervening mechanism (e.g. a longstanding sense of obligation to another individual in well-bonded relationships). In the latter case, bonding is a state built up over an extended period of time, with the level of bonding reflecting both the frequency of interaction and the duration of the relationship [51]. We test between these two possibilities in a set of two experiments that seek to establish three key points: (1) whether laughter enhances the sense of social bonding, (2) whether laughter increases the level of generosity towards others, and (3) whether an increase in generosity towards strangers, in particular, is mediated via the endorphin system or a sense of social bonding, or both. We use the Dictator Game as our index of prosociality (generosity) because we are explicitly interested in altruism, not trust (which underpins most of the public good games used in micro-economic experiments). We follow common practice [40, 41] in using changes in pain threshold as a proxy for endorphin activation [52]. Because the genders differ strikingly in both sensitivity to pain [53] and the psychological mechanisms that underpin friendship [54–56], we analyse the data separately for the two genders.

## Experiment 1

Experiment 1 aimed to determine whether laughter influenced generosity via the endorphin system. We test three hypotheses: that, relative to those in a control condition, individuals who watch a comedy video (1) laugh more often, and therefore (2) exhibit higher pain thresholds, in consequence of which (3) they are more generous in their donations to other group members. To be certain that any such effects are due to the endorphin system and not simply to changes in emotional state [57, 58], we also tested for correlated changes in positive and negative affect.

### Methods

Subjects were solicited via University lists: those who responded were invited to attend an experimental session in the Department with a friend (of either gender–all by one came with a friend of the same gender, however). In total, 66 subjects took part in the experiment. However, one pair who opted not to declare their gender and one pair who failed to complete all the tasks were excluded from the analyses, leaving 21 men and 41 women (mean age 24.6±9.0 years). Subjects were randomly assigned to one of three conditions: watching a comedy video alone or in a group of four (comedy conditions) or watching a golfing instruction video alone (control condition). The videos lasted ~20 mins in each case. While subjects were assigned individually to both of the 'Alone' conditions, they were assigned as a pair to the 'Group' condition. The requirement to bring a friend proved challenging, since the whole group session

was lost if one person came to their appointed session alone. All subjects were paid a flat-rate fee of £5 for attending. Subjects were not aware of the purpose of the experiment.

Laughter while watching the video was recorded on a dictaphone hung from each subject's neck; whether or not they laughed was subsequently sampled from the audio record at 15-sec intervals by AF. To determine pain threshold before and after watching the video, we used a standard sphygmomanometer blood pressure cuff on the dominant arm, recording how long the subject could stand the pain when the cuff was inflated to 300 Hg [38, 40]. The pretext was to measure physiological state. Pain thresholds were determined with subjects isolated in separate rooms while completing pre- and post-experimental paperwork and rating tasks. The index of interest was not absolute pain threshold but *change* in threshold after watching the video. After completing these tasks, subjects were invited to take part in two successive Dictator Games in which they could notionally donate some or all of their fee in units of £1 to (1) themselves versus the friend they came with or (2) themselves versus an unnamed stranger, for a maximum combined outlay of £10. Subjects did this task in isolation, and the order of the two donation tasks was randomised.

To determine whether any of the behavioural responses were due to affect rather than the endorphin-related effects of laughter (the two are not the same [43]), subjects completed the PANAS affect scale [59] before and after watching the video. The PANAS scale indexes positive and negative affect using 10 positive and 10 negative questions concerned with current emotional feelings, each rated on a 5-point scale. Scores for each scale were summed. Subjects were also asked to rate how emotionally close they felt to the friend they had come with on a 1–10 analogue scale.

Since, inevitably, laughter never occurred in the control (neutral) condition (it was explicitly designed not to elicit laughter), we use $\chi^2$ to compare whether laughter occurs or not (as a binary choice) for each subject across conditions. For all other dependent variables, we first tested for homogeneity of error variance using Levene's test. Since in every case except negative affect (PANAS, p = 0.038), the null hypothesis was upheld, we use parametric analysis of variance to test for main effects due to condition and gender and, more importantly, interaction effects. In each case, we treat condition as a fixed factor and gender as a random variable. Note that, of the parametric tests, ANOVA is considered the most robust to departures from parametric assumptions, and in particular departures from the assumption of normality. All correlation tests are Spearman correlations. Whenever specific directional hypotheses are under consideration in the correlation tests, 1-tailed tests are appropriate because a significant negative correlation would be as much evidence against the hypothesis as a non-significant correlation. In all other cases, 2-tailed tests are used.

Ethical approval was provided under Departmental arrangements. All subjects were over 18 and provided informed consent on signature.

The data are given in online "*S1 Data*".

## Results

Fig 1A plots the frequency of laughter for the three conditions. For both genders, the frequency of laughter increases from zero in the control condition to around 25% of samples in the group comedy condition. To test for a difference, we rescaled laughter frequencies as a simple binary choice (0 = no laughter; 1 = at least one scan sample with laughter). The differences across conditions, as a whole and for each gender separately, were significant ($\chi^2$ = 40.5, df = 2, p<0.0001; for each gender: $\chi^2_{Women}$ = 26.4, df = 2, p<0.0001; $\chi^2_{Men}$ = 14.1, df = 2, p = 0.001). Note that the data in Fig 1A suggest that laughter is approximately four times more likely in groups of four than when viewing the *same* comedy video alone.

Fig 1B and 1C plot the change in positive and negative affect (PANAS) across the three conditions. Positive affect increased across the conditions for both genders, and negative affect

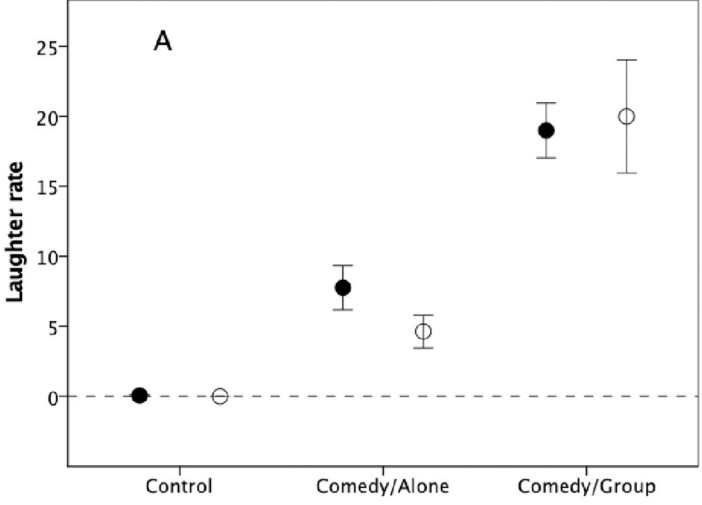

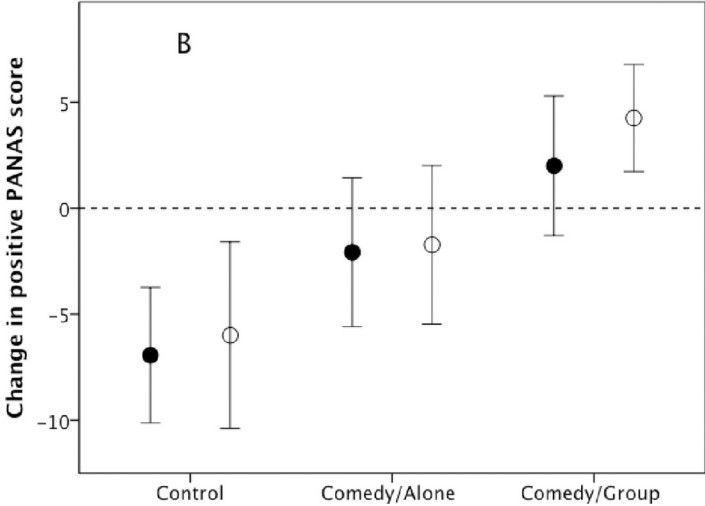

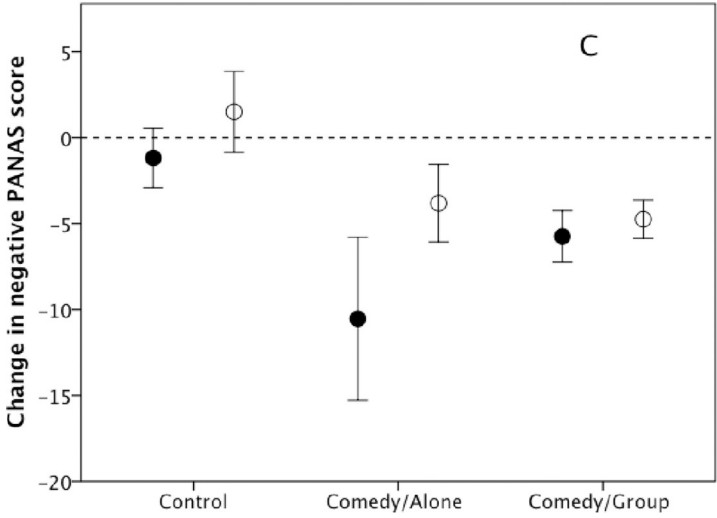

**Fig 1.** Mean (±se) (a) laughter rate (number of 15-sec scans on which subject was audibly laughing), (b) change in positive affect score and (c) change in negative affect score, for men (unfilled symbols) and women (filled symbols) in Experiment 1.

declined slightly in the two comedy conditions. However, only the increase in positive affect is significant (positive affect: condition, $F_{2,56} = 91.5$, p = 0.011, partial $\eta^2 = 0.989$; gender, $F_{1,56} = 1.2$, p = 0.194; condition*gender interaction, $F_{2,56} = 0.03$, p = 0.975; negative affect: condition, $F_{2,56} = 7.1$, p = 0.124; gender, $F_{1,56} = 3.6$, p = 0.177; condition*gender interaction, $F_{2,52} = 0.4$, p = 0.661). On an individual subject basis, the frequency of laughter was significantly positively correlated with the change in positive affect (Spearman $r_S = 0.434$, N = 62, p = 0.004) and negatively correlated with the change in negative affect ($r_S = -0.356$, N = 62, p = 0.003) (1-tailed tests in both cases since opposite-sign results would not be expected).

Fig 2A plots the change in pain threshold. The data suggest a more muted response in women, but a steep linear increase in men across the three conditions. Analysis of variance yields non-significant main effects for both condition ($F_{2,56} = 1.04$, p = 0.491) and gender ($F_{1,56} = 0.19$, p = 0.706), but a significant condition*gender interaction ($F_{2,56} = 5.33$, p = 0.008, partial $\eta^2 = 0.160$). This can be explained by the fact that, on the individual subject level, the frequency of laughter correlates significantly with the change in pain threshold in men ($r_S = 0.50$, N = 21, p = 0.010 1-tailed positive) but not in women ($r_S = -0.140$, N = 37, p = 0.705 1-tailed in a positive direction). Fig 2B plots the total amount donated in the two Dictator Games. There were no effects due to condition ($F_{2,54} = 0.61$, p = 0.621) or gender ($F_{1,54} = 0.34$, p = 0.612), and no interaction effect ($F_{2,54} = 0.39$, p = 0.677).

There was no correlation between total donation rate and pain threshold change in either gender (men: $r_S = 0.06$, N = 19, p = 0.411; women: $r_S = -0.002$, N = 41, p = 0.504; 1-tailed tests in both cases). Nor does donation rate correlate with laughter rate (women: $r_S = 0.17$, N = 37, p = 0.165; men: $r_S = -0.05$, N = 19, p = 0.582; all tests 1-tailed). Neither of the PANAS affect indices correlates significantly with changes in pain threshold (positive: $r_S = 0.05$, p = 0.687; negative: $r_S = -0.12$, p = 0.349) or with the amount of money donated (positive: $r_S = -0.03$, p = 0.832; negative: $r_S = 0.01$, p = 0.933) (N = 60, all tests 2-tailed, since no predictions are made). There was no correlation between subjects' rating of how emotionally close they were to their friend and the money they donated to the friend (women: $r_S = 0.10$, N = 41, p = 0.277; men: $r_S = 0.02$, N = 19, p = 0.467; 1-tailed tests) or their total donation to the group as a whole (women: $r_S = 0.10$, N = 41, p = 0.259; men: $r_S = 0.12$, N = 19, p = 0.309; 1-tailed tests). More importantly, perhaps, there was no correlation in the Comedy/Group condition (the only condition in which friends were together) in the donation scores of pairs of friends ($r_S = -0.012$, N = 8, p = 0.998 1-tailed).

Fig 3 plots the ratio of donation to Friend vs Self and to Stranger vs Friend as a function of experimental condition. The ratios do not vary significantly across condition or gender, and there is no interaction effect (Friend/Self: condition: $F_{2,38} = 1.38$, p = 0.421; gender: $F_{1,38} = 1.81$, p = 0.454; condition*gender interaction: $F_{2,38} = 0.27$, p = 0.762; Stranger/Friend: condition: $F_{2,49} = 0.03$, p = 0.973; 0; gender: $F_{1,49} = 0.49$, p = 0.778; condition*gender interaction: $F_{2,49} = 0.25$, p = 0.778). Note that, while subjects split their earnings more or less 50:50 with the friend, they were much less generous to the stranger (one-sample t-tests of $H_0$[ratio = 1] in each case: Friend/Self: $t_{43} = 0.64$, p = 0.524; Stranger/Friend: $t_{54} = -7.97$, p<0.0001).

## Discussion

Experiment 1 confirms previous studies that laughter triggered by comedy elevates pain thresholds [38], indicating activation of the endorphin system [39]. The results also confirm Provine's [60] claim that people are much more likely to laugh at something when in a group

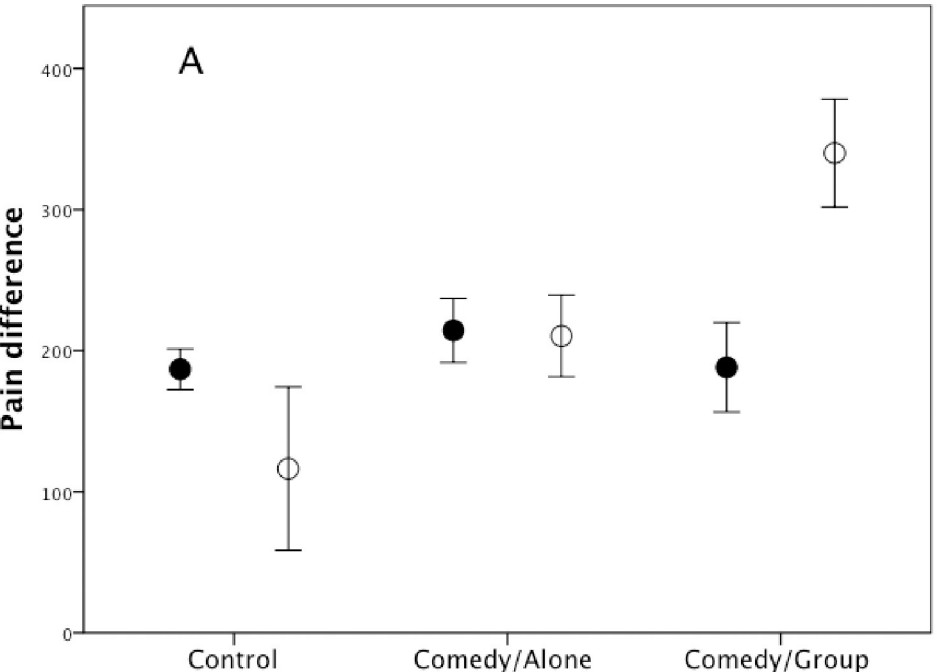

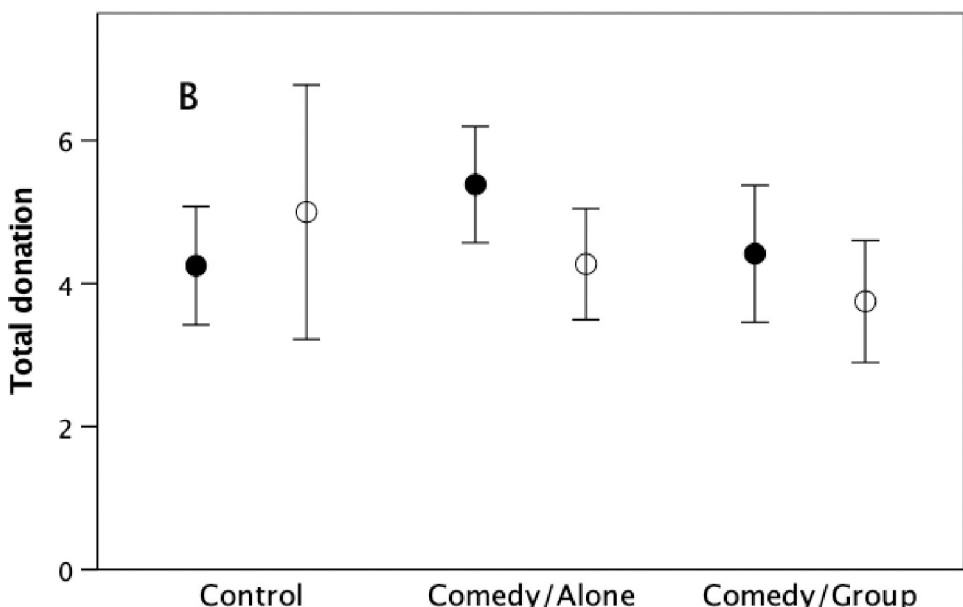

**Fig 2.** Mean±se (a) Change in pain threshold and (b) total donations, for men (unfilled symbols) and women (filled symbols) in Experiment 1.

than when they are alone. Provine simply noted whether or not someone laughed, whereas we estimated the proportion of time they spent laughing over a period of time, which not surprisingly yields a lower figure. Although there was an increase in positive affect and a reduction in

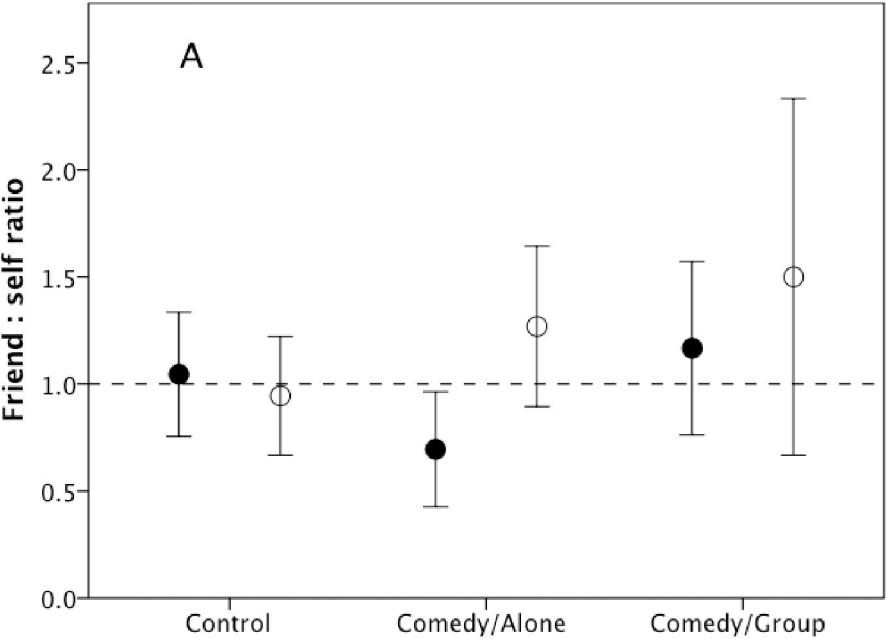

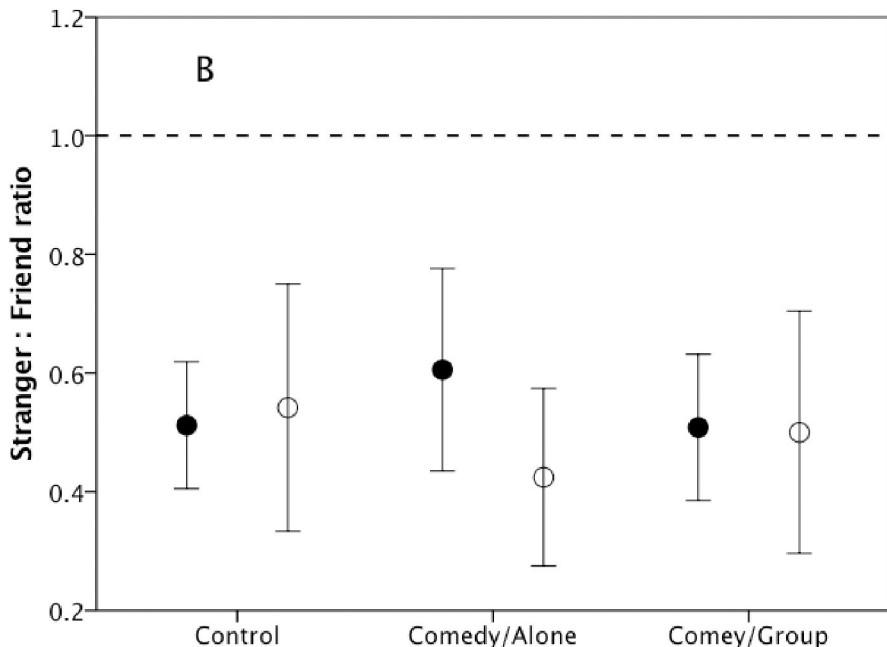

**Fig 3.** Mean±se ratio of (a) donation to friend divided by donation to self and (b) donation to stranger divided by donation to friend, for men (unfilled symbols) and women (filled symbols) in Experiment 1.

negative affect following laughter, the magnitude of these effects were, at best, modest (and not significant), suggesting that affect cannot explain the results [38, 43, 61]. Donations did not increase with laughter or pain threshold. Subjects were more generous to friends than to strangers while not differentiating between themselves and a close friend (as has been found in

previous experimental tests of Hamilton's Rule [8]). In some cases, there were significant differences between the genders in their responses, as would be expected given the gender differences in both sensitivity to pain [53] and the psychological mechanisms that underpin friendship [55, 56].

The main result of interest is the lack of any robust correlation between donations and either laughter or change in pain threshold. There are three possible explanations for this. One is that subjects were asked to make notional donations to both friends and strangers in the same game: generosity to friends might have influenced attitudes to strangers and levelled up donations, or *vice versa*. A second possibility is that donations depend not on laughter (and the endorphin system) *per se* but on the perceived degree of bonding to other group individuals, and that bonding is a consequence of a long term relationship rather than an immediate response to some behaviour or physiological response. Although there was no correlation between money donated to the friend and the self-rated emotional closeness to the friend, our experimental manipulation may not have been sufficient to trigger the required level of bonding on the day to affect donation rates to strangers. The third possibility is that altruism (donations) and sociality are two separate processes, and only one of them depends on the endorphin system. We know from other experiments that triggering the endorphin system increases the sense of bonding to the group [40, 41, 43, 48, 62]. This would, therefore, imply that altruism depends on other motivations (e.g. a sense of fairness, socially imposed rules) that are not influenced by laughter. Experiment 2 sought to test between these alternatives.

## Experiment 2

Experiment 2 aimed (1) to confirm that laughter results in elevated pain thresholds in groups when these consist only of strangers, (2) to test the new hypothesis that laughter increases the sense of bonding to strangers and (3) to determine whether donation rates were influenced by laughter, endorphin upregulation (indexed by a change in pain threshold) or bonding. Of the three possible explanations for the results in Experiment 1, we test the first by ensuring that all participants were strangers. If generosity to friends influenced attitudes towards strangers involved in the same experiment, then a stranger-only condition should remove that influence and allow a natural response to emerge. We tested the second explanation by including a psychological bonding measure so as to test directly whether laughter caused a change in bonding across the experimental manipulation. Finally, we tested the third possibility by asking whether donations were predicted by laughter, pain or bonding.

### Methods

Subjects were solicited from the Departmental subject panel, and were invited to attend an experimental session at a University location in preassigned groups of three or four. There was no overlap in the subject pool with Experiment 1. Experiment 2 followed the same design as Experiment 1 except that there were only two conditions: an experimental comedy condition using the same comedy video and a control condition (a TV wildlife documentary). The videos were 9m13s long in each case. 52 subjects (22 men; mean age 21.3 years) were randomly assigned to the two conditions. Care was taken to ensure that individuals did not know other group members, so that groups consisted of strangers. One subject did not complete all the tasks, and two subjects asked not to carry out the pain threshold task. Subjects received a £5 appearance fee. As before, subjects were not aware of the purpose of the experiment.

Since audio recordings made it difficult to pick up 'silent' laughs, subjects were videoed while watching the video and the number of laughter bouts, including broad smiles not accompanied by vocalisations, was subsequently coded by FG. The mean frequency of laughter so

defined in the comedy condition was 4.2±0.23 bouts per minute (range 2.8–6.9). As in Experiment 1, there was no laughter at all in the control condition. Pain thresholds were determined before and after watching the video using the wall-sit (or Roman Chair) task [8, 41, 63], which is less susceptible to subjective bias than the pressure cuff test. For this task, subjects adopt a seated position with their back against a wall and hips and knees bent at right angles, and are asked to hold the position for as long as possible until they collapse. The task is a standard skiing exercise. It is initially very comfortable, but becomes exponentially painful after about 30 secs; few people can maintain the position for more than about a minute. Subjects were tested in their groups on this task, since past experience with this task indicates that the presence of others makes little difference to how long the position is held. As with the blood pressure task, we are concerned only with the magnitude of the change in performance either side of an activity.

As a measure of bonding, subjects completed the *Inclusion-of-Other-in-Self* (IOS) task [64] with respect to the other members of their group before and after watching the video [40, 48]. The IOS task presents subjects with a set of seven pairs of circles that vary between almost complete overlap and no overlap at all in what is effectively a 7-point Likert scale; subjects are asked to specify which pair of circles best represented their emotional closeness to the other members of their experimental group. After completing these tasks, subjects were invited to decide what percentage of their £5 participation fee they would be willing to donate to the person on their immediate left in their group (subjects completed this part of the experiment seated in a circle, and were instructed not to discuss their answers).

Ethical approval was provided by the University of Oxford Combined University Research Ethics Committee (CUREC) under departmental provisions. All subjects were over 18 and provided written informed consent.

The data are given in online "*S2 Data*".

## Results

Fig 4A plots the frequency of laughter in the two conditions. As in Experiment 1, subjects were not expected to laugh in the control condition, so we rescaled rates of laughter to 0 (no laughter at all) or 1 (some laughter), and used a $\chi^2$ test to test for differences in laughter between conditions. There was significantly more laughter in the comedy condition ($\chi^2 = 25.0$, df = 1, p<0.0001), and this was true for both genders separately, though less strongly so for men ($\chi^2_{Women} = 25.2$, df = 1, p<0.0001; $\chi^2_{Men} = 4.1$, df = 1, p = 0.044). Fig 4B plots the equivalent values for change in pain threshold. There is a significant effect of condition ($F_{1,45} = 1584.0$, p = 0.016, partial $\eta^2 = 0.999$), with a marginal effect due to gender ($F_{1,45} = 125.4$, p = 0.057), but no condition*gender interaction ($F_{1,45} = 0.01$, p = 0.945).

Fig 5A plots change in IOS, or bonding index, across the experiment (rating after minus rating before). There are no main effects due to condition ($F_{1,47} = 0.7$, p = 0.549) or gender ($F_{1,47} = 0.01$, p = 0.932), but there is a significant condition*gender interaction effect ($F_{1,47} = 13.28$, p = 0.001, partial $\eta^2 = 0.220$) with women exhibiting an increase between conditions and men showing a much smaller change. The value of donations (Fig 5B) yielded a significant difference across conditions ($F_{1,47} = 187.0$, p = 0.046, partial $\eta^2 = 0.995$), but no main effect for gender ($F_{1,47} = 0.5$, p = 0.611) and no condition*gender interaction ($F_{1,47} = 0.03$, p = 0.866).

There are significant correlations between laughter rate and both changes in pain threshold (Spearman $r_S = 0.267$, p = 0.032) and the IOS bonding index ($r_S = 0.452$, p = 0.001), but only IOS correlates with the level of donation (laughter: $r_S = 0.141$, p = 0.163; pain threshold: $r_S = 0.053$, p = 0.358; IOS: $r_S = 0.425$, p = 0.036) (all tests 1-tailed for positive correlations, N = 49 subjects that completed all tasks). While laughter correlated positively (but weakly) with pain

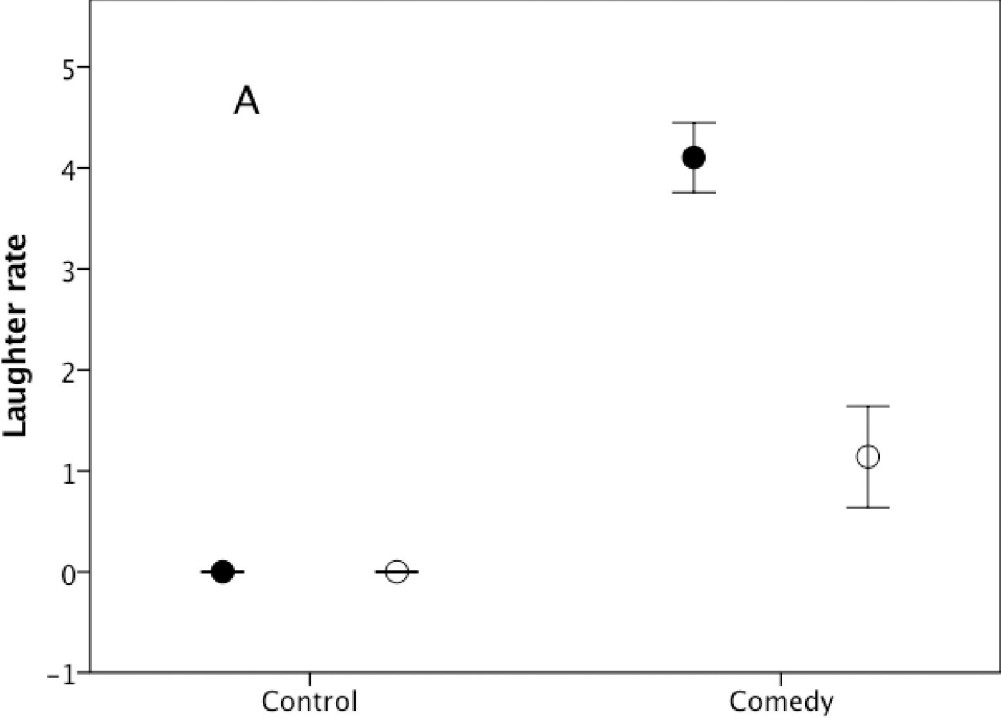

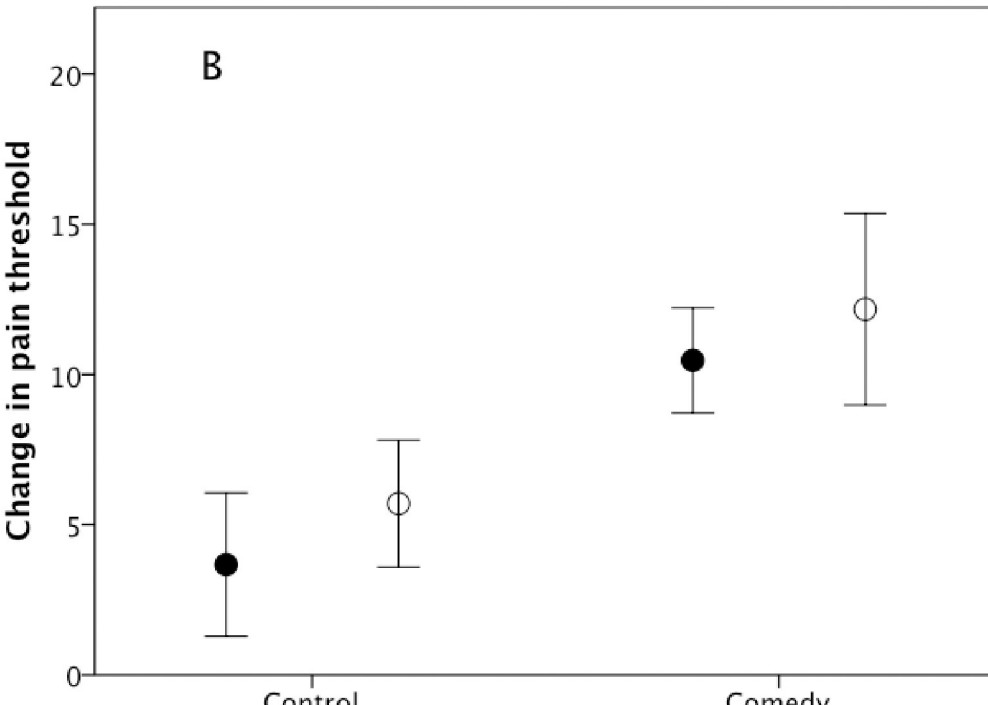

**Fig 4.** Mean±se (a) frequency of laughter (number of laughs given) and (b) change in pain threshold, for men (unfilled symbols) and women (filled symbols) in Experiment 2.

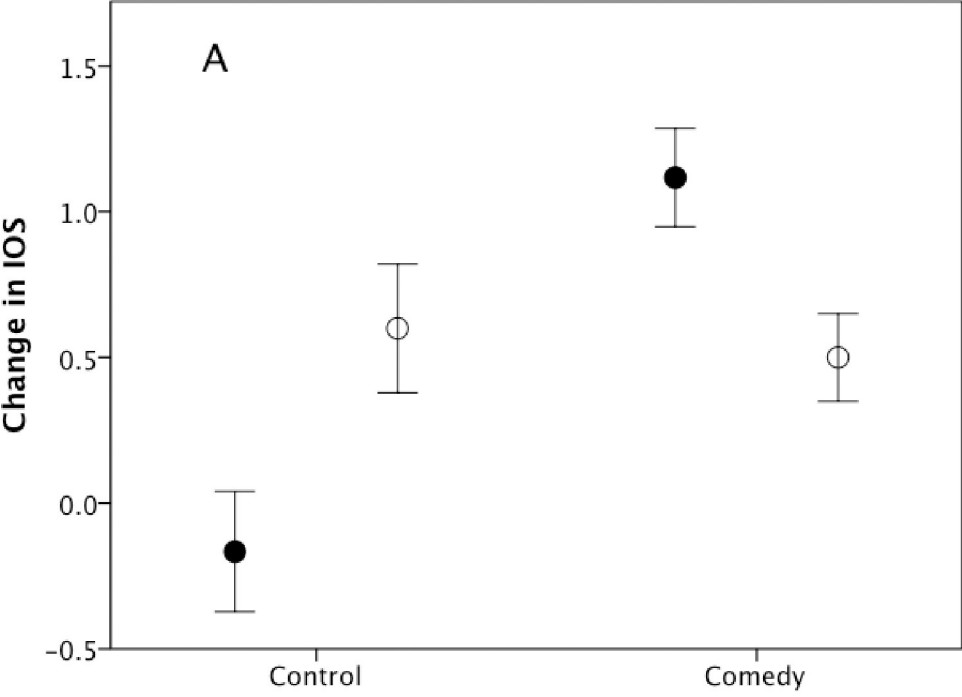

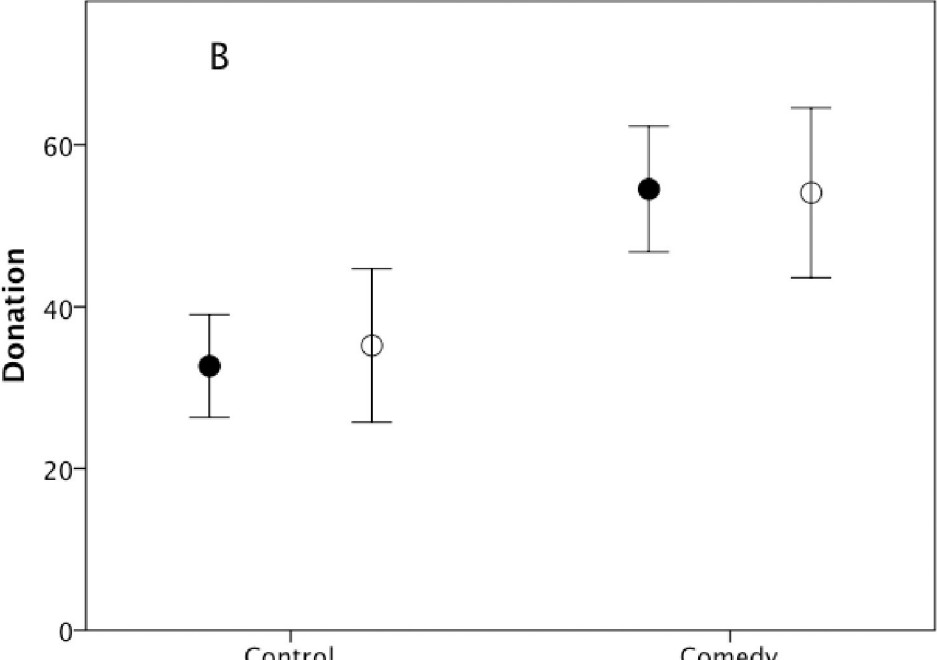

**Fig 5.** Mean±se (a) change IOS score (with respect to rest of experimental group) and (b) donation, for men (unfilled symbols) and women (filled symbols) in Experiment 2.

threshold change in both sexes (men, $r_S = 0.302$, N = 22, p = 0.086; women, $r_S = 0.303$, N = 27, p = 0.062), only women exhibited a significant positive correlation between laughter rate and IOS (men, $r_S = -0.021$, N = 22, p = 0.537; women, $r_S = 0.623$, N = 29, p<0.001). For women,

but not men, there was a weak (but only marginally significant) correlation between laughter rate and donations (men, $r_S$ = -0.163, N = 22, p = 0.766; women, $r_S$ = 0.288, N = 29, p = 0.065), and between IOS and donations (men, $r_S$ = 0.229, N = 22, p = 0.153; women, $r_S$ = 0.282, N = 29, p = 0.069). For neither gender, did pain threshold change correlate with donations (men, $r_S$ = 0.001, N = 22, p = 0.234; women, $r_S$ = 0.106, N = 27, p = 0.299). However, only the relationship for laughter with pain threshold and (for women) with IOS would survive correction for multiple testing. Indeed, a multiple regression with donation as the outcome variable and laughter rate, change in pain threshold and change in IOS as predictor variables was not significant, either overall ($F_{3,45}$ = 1.30, p = 0.287) or for the two genders individually (men: $F_{3,18}$ = 0.50, p = 0.690; women: $F_{3,23}$ = 1.39, p = 0.272), with no individual variables significant.

## Discussion

Experiment 2 confirmed the main finding from Experiment 1 that laughter induces a change in pain threshold (implying activation of the brain's endorphin system). More importantly, it also confirmed that there is an associated change in the sense of bonding to the group as a consequence of laughter. However, neither change in pain threshold nor change in perceived bonding correlated at all strongly with level of generosity to strangers. The fact that men did not exhibit as strong a bonding response as women (Fig 5A) might have been due to their much lower rates of laughter when watching the comedy video (Fig 4A); alternatively, it may simply reflect the fact that men's friendships are more casual and less intense than those of women [54, 65].

Rates of laughter were higher in Experiment 2 than in Experiment 1 for women but not men, but it is not possible to say whether this was due to an increase in audible or inaudible laughter. However, the difference between the experiments in the laughter rates of the two genders would not explain the patterns for either pain threshold change or donation rate between the two experiments.

Of the three alternative explanations for the Experiment 1 results, the first is excluded by the experimental design (friends are excluded) and the second (that it is bonding, not laughter, that determines donation level) does not yield a significant effect. The third possibility (that bonding and prosociality are separate processes underpinned by different (neuro-)psychological mechanisms) is supported by the lack of correlation between donation and any of the other three variables, making this the most likely explanation.

## General discussion

Taken together, these results confirm that laughter elevates pain thresholds, a proxy for activation of the brain's endorphin system [39], and that laughter is also associated with an increased sense of bonding to strangers (at least in women). Although laughter resulted in greater generosity to strangers in Experiment 2, the significance level was on the margin (p≈0.05), there was no effect in Experiment 1; moreover, in neither experiment was there was a quantitative relationship between the amount of laughter and the size of donation. More importantly, change in pain threshold (indicating endorphin upregulation) did not correlate with size of donation in either experiment. The data in Fig 3 confirm that generosity to friends is much greater than generosity to strangers [8], suggesting that longstanding bonds trump any instantaneous bonds created by laughter. Experiment 1 allowed us to rule out affect as a likely explanation for these results. Given that laughter has been shown to upregulate the brain's endorphin system [39] and that both endorphin receptor density [21] and baseline pain threshold [66] correlate with social engagement (indexed by attachment style and number of friends, respectively), Experiment 2 allows us to conclude that the relationship between

laughter and enhanced bonding reported here is mediated by the brain's endorphin system, whereas generosity (at least to strangers) is not, suggesting that bonding and prosocial generosity may be underpinned by two different psychological mechanisms.

The suggestion that two mechanisms may be involved is reinforced by studies that have used behavioural synchrony to induce a sense of bonding. Sullivan et al. [67] showed that behavioural synchrony in a physical exercise (a phenomenon known to enhance sense of bonding [41, 48] directly influenced willingness to cooperate with other group members on a public good game, but that this was not mediated by a change in pain threshold. Using a similar experimental design, Lang et al. [68] found that engaging in a highly synchronised physical exercise increased both the perceived likeability of the confederate with whom they interacted and willingness to invest in a Trust game with them; however, endorphin upregulation (indexed as pain threshold change) mediated only performance on the investment game. Note that to assess generosity, both these studies used a standard economic Public Good Game, which has the form of a trust task (subjects invest in a central pool in the expectation that everyone else will do so, otherwise they risk losing their capital and would have done better to hold onto their money). When played repeatedly against strangers, such tasks invariably exhibit a decline of investments into the common pool with successive trials [69, 70] as individuals become increasingly aware that they can benefit by behaving selfishly.

Thus, despite using a variety of different ways of triggering the endorphin system and several different kinds of prosociality tasks, these studies and ours collectively seem to suggest that bondedness and prosociality may be underpinned by different mechanisms. We suggest that this is because cooperative tasks, or tasks that are essentially forms of economic trading, are not central to the dynamics of core human social groups (i.e. personal social networks), even though they may provide the principal function for very large scale groupings (e.g. tribes in traditional societies) [71]. Personal social networks, like all small scale communities, exist to provide mutual support and ensure coordination rather than trading arrangements, and are based on bonding and a sense of obligation rather than economic exchange [36, 72]. At the larger social scale beyond this, most individuals are to all intents and purposes strangers [72], and the rules of trading are those of commerce and not those of community. Such arrangements will be more open to the establishment of cultural norms [10].

Endorphin-based social bonding may require repeated interactions to build into a fully bonded relationship that leads to unconditional prosociality (Fig 3). In contrast, instantaneous bonding to a group of strangers may not, of itself, make a transactional relationship with a stranger more likely. In other words, it is only when the bond reaches a specific strength that unconditional prosociality becomes the norm in a dyadic relationship [8, 51]. This may make the use of conventional economic games inappropriate as tools for exploring human *social* (as opposed to economic) behavior [10, 70]. In other words, there is an important distinction between bonded relationships (as an intermediate step to support unconditional cooperation) and purely transactional (or trading) relationships. In short, human (and anthropoid primate [26, 27]) sociality involves personalised relationships of obligation and trust, but not necessarily one-off transactional relationships where different rules of trade and fairness may apply.

Although the effects we found appear to be robust (and are confirmed by studies using different designs), we should note that they depend on using changes in pain threshold as a proxy for endorphin activation. While there is extensive evidence from pain research to justify this [52, 53, 73, 74], a more direct test for endorphin activation would obviously provide stronger confirmation. Such tests are not, however, straightforward because endorphins do not cross the blood-brain barrier [75, 76]. CNS endorphin assays require either lumbar puncture or PET scanning, neither of which are pleasant to experience or free of side effects. Pain threshold assays are generally considered the better option, although the use of endorphin antagonists

(e.g. naltrexone) may provide indirect evidence for the role of endorphins [41]. A second issue is that are all our participants were students. A replication with older subjects, a wider ethnic range, and larger samples would be desirable. None of these, of course, are novel limitations, but they do remind us that experiments are always subject to practical constraints that may limit their generalizability until they have been replicated. Our limited range of videos might also seem a weakness of experimental design. However, we are not here testing the *causes* of laughter; rather, we are testing the *consequences* of laughter. So long as our subjects laugh, we have no interest in why they laugh (i.e. whether some videos are funnier than others). In any case, we should note that other studies of laughter [38] have used a wider variety of comedy (and control) videos, and all comedy videos stimulate laughter.

A more serious weakness of our design is the fact that subjects made purely notional decisions about how to allocate their endowment. Their decisions might have been very different if they actually had to give their money away to a stranger. In mitigation, we merely note that, in previous studies where subjects have been asked to earn money for others [8], the results have been broadly similar to those reported here. More importantly, given that people invariably claim to be *more* generous than they actually are [77], this could only weaken the relationship between laughter and generosity and thus reinforce our claim that prosociality is underpinned by a different mechanism.

In sum, we show that, like a number other specifically social human behaviours (including singing, dancing and feasting [40, 48, 49]), laughter promotes social bonding in a group, and can do so even among complete strangers (although this may be true mainly for women). Unlike these other behaviours, laughter is something we share with the great apes [78], and is likely to have deep evolutionary and biological roots; in contrast, the other behaviours have more recent historical origins. Perhaps because of this, generosity (prosociality) towards strangers is unrelated either to laughter and the endorphin flux it generates or to the momentary psychological sense of bonding that these produce. Rather, altruistic acts of this kind seem to involve different psychological or cultural mechanisms from those designed to create bonded social relationships and bonded groups.

## Supporting information

**S1 Data. Study 1.**
(XLSX)

**S2 Data. Study 2.**
(XLSX)

## Author Contributions

**Conceptualization:** R. I. M. Dunbar, Felix Grainger, Eiluned Pearce.

**Formal analysis:** R. I. M. Dunbar, Felix Grainger.

**Funding acquisition:** R. I. M. Dunbar.

**Investigation:** Anna Frangou, Felix Grainger, Eiluned Pearce.

**Methodology:** R. I. M. Dunbar, Anna Frangou, Felix Grainger.

**Project administration:** R. I. M. Dunbar, Anna Frangou.

**Supervision:** R. I. M. Dunbar.

**Writing – original draft:** R. I. M. Dunbar.

**Writing – review & editing:** R. I. M. Dunbar, Anna Frangou, Felix Grainger, Eiluned Pearce.

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
