## [Decision Letter · Decision Letter 0]

10 Jun 2021

PONE-D-21-08705

Laughter, Social Bonding, and the Generosity of Friends and Strangers

PLOS ONE

Dear Dr. Dunbar,

Thank you for submitting your manuscript to PLOS ONE. After careful consideration, we feel that it has merit but does not fully meet PLOS ONE’s publication criteria as it currently stands. Therefore, we invite you to submit a revised version of the manuscript that addresses the points raised during the review process.

Please find below the reviewers' comments, as well as those of mine.

We look forward to receiving your revised manuscript.

Kind regards,

Valerio Capraro

Academic Editor

PLOS ONE

Journal Requirements:

2. Please change "female” or "male" to "woman” or "man" as appropriate, when used as a noun (see for instance https://apastyle.apa.org/style-grammar-guidelines/bias-free-language/gender).

3. Please modify the title to ensure that it is meeting PLOS’ guidelines (https://journals.plos.org/plosone/s/submission-guidelines#loc-title). In particular, the title should be "specific, descriptive, concise, and comprehensible to readers outside the field" and in this case it is not sufficiently informative and specific about your study's scope and methodology.

Additional Editor Comments:

I have now collected two reviews from two experts in the field, whom I thank for their punctual, detailed, and constructive feedback. Both reviewers think that the topic of the paper is interesting, but their final judgment is very split, with one recommending minor revision and one recommending rejection. Looking at the negative review, I find that the main criticisms regard a lack of connection of this work with previous related work, a lack of clarity in the experimental design of Study 1 and probably an overstatement of the results of the second experiment. I think that these points can potentially be addressed with a suitable revision. Therefore, I have decided to invite you to revise your work for Plos One. Needless to say that all reviewers' comments, from both reviewers, should be addressed.

I am looking forward for the revision.

Reviewers' comments:

Reviewer's Responses to Questions

**Comments to the Author**

1. Is the manuscript technically sound, and do the data support the conclusions?

Reviewer #1: No

Reviewer #2: Yes

2. Has the statistical analysis been performed appropriately and rigorously? 

Reviewer #1: Yes

Reviewer #2: Yes

3. Have the authors made all data underlying the findings in their manuscript fully available?

Reviewer #1: Yes

Reviewer #2: Yes

4. Is the manuscript presented in an intelligible fashion and written in standard English?

Reviewer #1: No

Reviewer #2: Yes

5. Review Comments to the Author

Reviewer #1: This study tries to answer two questions: Does laughter increases social bonding, and does social bonding in return increase pro-sociality? (see abstract). Both questions are very interesting. However, the authors do not link their research sufficiently to the already existing evidence. For instance, on page 4 they write: “While it may seem obvious that a sense of social bonding and altruism ought to be associated, this key relationship has never been directly tested. However, most studies of cooperation and altruism have used economic games between strangers, ignoring the fact that cooperation and altruism occurs mainly between individuals who have explicit relationships with each other or belong to the same small community.” This is an inadequate description of the literature in experimental game theory. There are numerous studies investigating the effect of social distance on cooperation and prosocial behavior. I provide just a few examples:

1) Goeree et al. (2010): The 1/d Law of Giving. American Economic Journal: Microeconomics 2: 183–203.

2) Engel (2011): Dictator games: a meta study. Experimental Economics 14: 583–610.

3) Candelo et al. (2018) Social Distance Matters in Dictator Games: Evidence from 11 Mexican Villages. Games 9,77.

I am also confused about the design of the first experiment. On page 5 the authors write “We use an experimental design to test (1) whether laughter increases the level of generosity towards others, (2) whether laughter enhances the sense of social bonding, and (3) whether an increase in generosity towards strangers, in particular, is mediated via the endorphin system or a sense of social bonding, or both.” But experiment 1 does not include a measure of social bonding, but looks only at the relation of laughter on generosity. However, laughing should only increase social bonding in the group condition, to those one is laughing with. How this was taken into account in the matching of the recipients in the dictator game remains unclear. Overall, my impression is that experiment 1 does help to answer the questions raised by the authors. It introduces confusion and makes it harder to follow the line of reasoning in the manuscript.

The second experiment addresses many problems of experiment 1 and is much better linked to the research questions. However, the results are inconclusive. For instance, the finding that laughter elevates bonding is only found for females. This fact is downplayed in the general discussion when the authors write “Taken together, these results confirm that laughter elevates pain thresholds, a proxy for activation of the brain’s endorphin system (Manninen et al. 2017), and suggest that laughter is also associated with an increased sense of bonding to strangers (albeit more strongly in females than males).” Similarly, on page 18: “In sum, we show that, like a number other specifically social human behaviours (…), laughter promotes social bonding in a group, and can do so even among complete strangers.” In fact, it is only found in females in experiment 2. Peculiarly, this important caveat is not discussed by the authors, and their general conclusion is only partly supported by the results of experiment 2.

In sum, I am not convinced that the manuscript makes an important contribution to the literature on bonding and giving in dictator games.

Reviewer #2: This paper studies the relations between laughter, tolerance to pain, bonding, and generosity. In two experimental studies, the authors manipulate participants’ exposure to a comedy video (which successfully induced laughter) and then measure laughter (using audio and video recording), tolerance to pain (via a pressure-cuff task in Exp. 1 and a wall-sit task in Exp. 2), emotional closeness and bonding, as well as generosity in hypothetical decision-making tasks (using dictator games). They find that laughter is associated with higher pain thresholds (which they interpret as consistent with the activation of the endorphin system), and with a stronger sense of bonding to strangers (especially among women, see Exp. 2), but that induced laughter is not associated with increased generosity (i.e., donations in the dictator game).

Overall, this is an interesting and clearly written manuscript. I agree with the authors that it is important to study prosocial behaviors in a relational context, looking into interactions between different relationship partners (here, both friends and strangers). I think the reported experiments can provide insights into social bonding and generosity—and the role of laughter in promoting one versus the other. That said, I have some questions and recommendations regarding the methods and analyses used, as well as regarding the interpretation of results.

1. As the authors note in the discussion, the sample sizes of the two experiments are relatively small. It could be then that the studies are underpowered to detect some effects of interest. I would recommend that the authors report how they determined the targeted sample size for the studies. Further, the authors can consider conducting sensitivity analyses to give readers some idea of the smallest effect size that their studies would be able to detect, given their sample sizes.

2. It is not clear to me why gender is included as a predictor in all main analyses. Are there a priori reasons to expect that men and women will differ in the DVs and, perhaps more importantly, that gender would moderate the effects of interest? Presumably, presenting the main analyses without the gender main effects and interactions would not change results by much, but could make for a clearer presentation of main findings.

3. One reason that the authors observe no effects of laughter on generosity could be that they use hypothetical dictator games. Given that decisions are hypothetical, there is no cost associated with being generous. It seems that indeed participants generally follow equality norms, donating about half of their endowment across conditions. The authors acknowledge this issue in the discussion and suggest that “asking people to actually give away their money might be expected to produce a less generous outcome than the one we observed.” But that might be exactly the situation in which laughter, endorphin activation, and bonding could make a difference. If decisions are costly, having experienced something funny and having bonded with others could make one more willing to pay the cost and be generous (than if no common bonding experience has occurred).

4. In the second experiment, the authors use videos, based on which they can code more nuanced expressions of laughter (e.g., smiles that are not audible). Have they perhaps tried to also use a similar coding of audible laughter as in Experiment 1? If so, are the occurrences of audible laughter similarly frequent across the experiments?

5. In the results of Experiment 2, the authors note that “none of these [the other variables of interest] correlate with the level of donation.” However, the results in parentheses suggest that there is a statistically significant and moderately strong association between bonding (IOS) and generosity.

6. In the discussion, the authors note that “it seems likely that the relationship between laughter and enhanced bonding is mediated by the brain’s endorphin system, whereas generosity to strangers is underpinned by a different mechanism.” My impression is that the experiments in the paper do not directly test this—i.e., a mediating role of endorphin activation and pain tolerance. Perhaps the authors could clarify whether this is speculative.

7. Finally, a related point: In the discussion, p. 15, the authors cite studies by Sullivan et al. (2015) and Lang et al. (2017) which have found evidence that mechanisms other than endorphin system activation might motivate generosity. Could the authors provide a few more details in text about what these alternative potential mechanisms might be?

6. PLOS authors have the option to publish the peer review history of their article (what does this mean?). If published, this will include your full peer review and any attached files.

Reviewer #1: No

Reviewer #2: No

---

## [Author Response · Author response to Decision Letter 0]

22 Jul 2021

RESPONSE: this has now been done. 

2. Please change "female” or "male" to "woman” or "man" as appropriate, when used as a noun (see for instance https://apastyle.apa.org/style-grammar-guidelines/bias-free-language/gender).

RESPONSE: this has now been done. 

3. Please modify the title to ensure that it is meeting PLOS’ guidelines (https://journals.plos.org/plosone/s/submission-guidelines#loc-title). In particular, the title should be "specific, descriptive, concise, and comprehensible to readers outside the field" and in this case it is not sufficiently informative and specific about your study's scope and methodology.

RESPONSE: this has now been done. 

RESPONSE: this has now been done. 

Additional Editor Comments:

I have now collected two reviews from two experts in the field, whom I thank for their punctual, detailed, and constructive feedback. Both reviewers think that the topic of the paper is interesting, but their final judgment is very split, with one recommending minor revision and one recommending rejection. Looking at the negative review, I find that the main criticisms regard a lack of connection of this work with previous related work, a lack of clarity in the experimental design of Study 1 and probably an overstatement of the results of the second experiment. I think that these points can potentially be addressed with a suitable revision. Therefore, I have decided to invite you to revise your work for Plos One. Needless to say that all reviewers' comments, from both reviewers, should be addressed.

I am looking forward for the revision.

Comments to the Author

Reviewer #1: 

This study tries to answer two questions: Does laughter increases social bonding, and does social bonding in return increase pro-sociality? (see abstract). Both questions are very interesting. However, the authors do not link their research sufficiently to the already existing evidence. For instance, on page 4 they write: “While it may seem obvious that a sense of social bonding and altruism ought to be associated, this key relationship has never been directly tested. However, most studies of cooperation and altruism have used economic games between strangers, ignoring the fact that cooperation and altruism occurs mainly between individuals who have explicit relationships with each other or belong to the same small community.” This is an inadequate description of the literature in experimental game theory. There are numerous studies investigating the effect of social distance on cooperation and prosocial behavior. I provide just a few examples:

1) Goeree et al. (2010): The 1/d Law of Giving. American Economic Journal: Microeconomics 2: 183–203.

2) Engel (2011): Dictator games: a meta study. Experimental Economics 14: 583–610.

3) Candelo et al. (2018) Social Distance Matters in Dictator Games: Evidence from 11 Mexican Villages. Games 9,77.

RESPONSE: We thank the reviewer for drawing our attention to these papers, which we now cite. We should note, however, that our paper is not concerned with testing whether social distance is a factor in giving but with whether laughter, acting via the endorphin system, facilitates giving. We raise the issue of social distance simply to point to something we need to consider as a confound. Our point (which remains true) is (a) that people discriminate between degrees of social closeness (as the papers we cite also show) and, in passing, (b) that this isn’t often taken into account in microeconomic experiments (as the Engel meta-analysis of the field itself confirms: only 1% of the subjects in his total sample took part in experiments where social distance was a measured variable). The text has been amended to reflect this more clearly.

I am also confused about the design of the first experiment. On page 5 the authors write “We use an experimental design to test (1) whether laughter increases the level of generosity towards others, (2) whether laughter enhances the sense of social bonding, and (3) whether an increase in generosity towards strangers, in particular, is mediated via the endorphin system or a sense of social bonding, or both.” But experiment 1 does not include a measure of social bonding, but looks only at the relation of laughter on generosity. However, laughing should only increase social bonding in the group condition, to those one is laughing with. How this was taken into account in the matching of the recipients in the dictator game remains unclear. Overall, my impression is that experiment 1 does help to answer the questions raised by the authors. It introduces confusion and makes it harder to follow the line of reasoning in the manuscript.

RESPONSE: The aims cited by the referee are not the aims of Experiment 1; they refer to the combined overall aims of the study. The aims of Experiment 1 are stated in the text as: “[those] who watch a comedy video (1) laugh more often, and therefore (2) exhibit higher pain thresholds, in consequence of which (3) they are more generous in their donations than those that watch a control video.” To avoid confusion, the aims of the study as a whole on p. 5 have been restructured to reflect the sequence of experiments.

The second experiment addresses many problems of experiment 1 and is much better linked to the research questions. However, the results are inconclusive. For instance, the finding that laughter elevates bonding is only found for females. This fact is downplayed in the general discussion when the authors write “Taken together, these results confirm that laughter elevates pain thresholds, a proxy for activation of the brain’s endorphin system (Manninen et al. 2017), and suggest that laughter is also associated with an increased sense of bonding to strangers (albeit more strongly in females than males).” 

RESPONSE: The results are not, in fact, inconclusive. They show that laughter does not trigger generosity via the endorphin system, but rather generosity has more to do with history of the relationship. In other words, that bonding and prosociality are two different things, acting via different routes. That is a very significant finding in the context of understanding social processes (even if it is disappointing from a purely economic point of view). We have revised the text to clarify this.

Similarly, on page 18: “In sum, we show that, like a number other specifically social human behaviours (…), laughter promotes social bonding in a group, and can do so even among complete strangers.” In fact, it is only found in females in experiment 2. Peculiarly, this important caveat is not discussed by the authors, and their general conclusion is only partly supported by the results of experiment 2.

RESPONSE: This has now been rectified.

In sum, I am not convinced that the manuscript makes an important contribution to the literature on bonding and giving in dictator games.

RESPONSE: We fear that this observation misses the point of the study: this study is not about micro-economic experiments, the Dictator Game or altruistic prosociality, but about whether laughter facilitates generosity. We know of no experiment that actually tests that claim – but would, of course, be glad to know of any that we have not come across in the economics literature.

Reviewer #2: 

This paper studies the relations between laughter, tolerance to pain, bonding, and generosity. In two experimental studies, the authors manipulate participants’ exposure to a comedy video (which successfully induced laughter) and then measure laughter (using audio and video recording), tolerance to pain (via a pressure-cuff task in Exp. 1 and a wall-sit task in Exp. 2), emotional closeness and bonding, as well as generosity in hypothetical decision-making tasks (using dictator games). They find that laughter is associated with higher pain thresholds (which they interpret as consistent with the activation of the endorphin system), and with a stronger sense of bonding to strangers (especially among women, see Exp. 2), but that induced laughter is not associated with increased generosity (i.e., donations in the dictator game).

Overall, this is an interesting and clearly written manuscript. I agree with the authors that it is important to study prosocial behaviors in a relational context, looking into interactions between different relationship partners (here, both friends and strangers). I think the reported experiments can provide insights into social bonding and generosity—and the role of laughter in promoting one versus the other. That said, I have some questions and recommendations regarding the methods and analyses used, as well as regarding the interpretation of results.

1. As the authors note in the discussion, the sample sizes of the two experiments are relatively small. It could be then that the studies are underpowered to detect some effects of interest. I would recommend that the authors report how they determined the targeted sample size for the studies. Further, the authors can consider conducting sensitivity analyses to give readers some idea of the smallest effect size that their studies would be able to detect, given their sample sizes.

RESPONSE: Part of the reason for the small sample size was given in the Methods: subjects had to come in pairs, and if one member failed to turn up the whole session had to be abandoned. In addition, it just proved difficult to recruit people, mainly due to the out-of-the-way location where we ran the experiment. For this reason, partial eta’s were given to indicate the magnitude of the effects in each case.

2. It is not clear to me why gender is included as a predictor in all main analyses. Are there a priori reasons to expect that men and women will differ in the DVs and, perhaps more importantly, that gender would moderate the effects of interest? Presumably, presenting the main analyses without the gender main effects and interactions would not change results by much, but could make for a clearer presentation of main findings.

RESPONSE: Gender was included because there are well-attested differences in (a) basal pain threshold and (b) sociality and prosocial demeanour. We have added detail on this to the Introduction.

3. One reason that the authors observe no effects of laughter on generosity could be that they use hypothetical dictator games. Given that decisions are hypothetical, there is no cost associated with being generous. It seems that indeed participants generally follow equality norms, donating about half of their endowment across conditions. The authors acknowledge this issue in the discussion and suggest that “asking people to actually give away their money might be expected to produce a less generous outcome than the one we observed.” But that might be exactly the situation in which laughter, endorphin activation, and bonding could make a difference. If decisions are costly, having experienced something funny and having bonded with others could make one more willing to pay the cost and be generous (than if no common bonding experience has occurred).

RESPONSE: The usual expectation (and finding) is that people are more generous in virtual/hypothetical situations than when they have to hand over actual money. This should mean that the differences between laughter and no-laughter conditions will be smaller than they already are. We add a comment to this effect to the Discussion.

4. In the second experiment, the authors use videos, based on which they can code more nuanced expressions of laughter (e.g., smiles that are not audible). Have they perhaps tried to also use a similar coding of audible laughter as in Experiment 1? If so, are the occurrences of audible laughter similarly frequent across the experiments?

RESPONSE: Rates of laughter in Expt 2 were similar to those in Expt 1 for males, but higher for females. We cannot now determine whether the latter was due to increased audible or inaudible laughter (the video recordings had to be destroyed after the initial data were extracted). This difference did not, however, relate to patterns of donation or pain difference. A note on this has been added to the text.

5. In the results of Experiment 2, the authors note that “none of these [the other variables of interest] correlate with the level of donation.” However, the results in parentheses suggest that there is a statistically significant and moderately strong association between bonding (IOS) and generosity.

RESPONSE: But, as stated immediately afterwards, this does not survive Bonferroni correction for multiple tests. We applied a precautionary principle here.

6. In the discussion, the authors note that “it seems likely that the relationship between laughter and enhanced bonding is mediated by the brain’s endorphin system, whereas generosity to strangers is underpinned by a different mechanism.” My impression is that the experiments in the paper do not directly test this—i.e., a mediating role of endorphin activation and pain tolerance. Perhaps the authors could clarify whether this is speculative.

RESPONSE: The point is a good one. However, while it is correct that we did not test directly whether endorphins mediate the relationship in this paper, we have established elsewhere both that laughter triggers the endorphin system [Manninen et al. 2017, J Neurosience] and that endorphin upregulation correlates with more intense bonding [Nummenmaa et al. 2015, Human Brain Mapping; Tarr et al. 2015, Evol Human Behav]. The Discussion has been adjusted to reflect this point.

7. Finally, a related point: In the discussion, p. 15, the authors cite studies by Sullivan et al. (2015) and Lang et al. (2017) which have found evidence that mechanisms other than endorphin system activation might motivate generosity. Could the authors provide a few more details in text about what these alternative potential mechanisms might be?

RESPONSE: Details now added.

---

## [Editor Report · Decision Letter 1]

3 Aug 2021

Laughter Influences Social Bonding But Not Prosocial Generosity to Friends and Strangers

PONE-D-21-08705R1

Dear Dr. Dunbar,

We’re pleased to inform you that your manuscript has been judged scientifically suitable for publication and will be formally accepted for publication once it meets all outstanding technical requirements.

Kind regards,

Valerio Capraro

Academic Editor

PLOS ONE
---

## [Editor Report · Acceptance letter]

5 Aug 2021

PONE-D-21-08705R1 

Laughter Influences Social Bonding But Not Prosocial Generosity to Friends and Strangers 

Dear Dr. Dunbar:

I'm pleased to inform you that your manuscript has been deemed suitable for publication in PLOS ONE. Congratulations! Your manuscript is now with our production department. 

Kind regards, 

on behalf of

Dr. Valerio Capraro 

Academic Editor

PLOS ONE